# Pro-Apoptotic Activity of 1-(4,5,6,7-Tetrabromo-1*H*-benzimidazol-1-yl)propan-2-one, an Intracellular Inhibitor of PIM-1 Kinase in Acute Lymphoblastic Leukemia and Breast Cancer Cells

**DOI:** 10.3390/ijms26125897

**Published:** 2025-06-19

**Authors:** Patrycja Wińska, Monika Wielechowska, Łukasz Milewski, Paweł Siedlecki, Edyta Łukowska-Chojnacka

**Affiliations:** 1Faculty of Chemistry, Warsaw University of Technology, Noakowskiego St. 3, 00-664 Warsaw, Poland; monika.wielechowska@pw.edu.pl (M.W.); edyta.chojnacka@pw.edu.pl (E.Ł.-C.); 2Institute of Biochemistry and Biophysics, Polish Academy of Sciences, Pawińskiego St. 5A, 02-106 Warsaw, Poland; lmilewski@ibb.waw.pl (Ł.M.); pawel@ibb.waw.pl (P.S.)

**Keywords:** TBBi derivatives, protein kinase CK2, PIM kinase, anti-tumor activity, leukemia, breast cancer

## Abstract

Inhibition of CK2 and/or PIM-1 kinases has been shown to induce apoptosis in a variety of cancer cell lines, underscoring their potential as valuable targets in anti-cancer drug development. In this study, a series of *N*-substituted derivatives of 4,5,6,7-tetrabromo-1*H*-benzimidazole, including 2-oxopropyl/2-oxobutyl substituents and their respective hydroxyl analogues, were synthesized and evaluated for anti-cancer activity. The compounds’ ability to inhibit CK2α and PIM-1 kinases was assessed through enzymatic assays, complemented by comprehensive in silico enzyme–substrate docking analyses. Cytotoxicity was evaluated using the MTT assay in human cancer cell lines—including acute lymphoblastic leukemia (CCRF-CEM) and breast cancer (MCF-7, MDA-MB-231)—as well as in normal Vero cells. Apoptosis induction in the two most responsive cell lines (CCRF-CEM and MCF-7) was further examined using flow cytometry-based assays, including annexin V binding, mitochondrial membrane potential disruption, caspase-3 activation, and cell cycle analysis. Intracellular inhibition of CK2 and PIM-1 kinases was confirmed in CCRF-CEM and MCF-7 cells using Western blot and phospho-flow cytometry. Among the synthesized compounds, we identified a novel TBBi derivative exhibiting pronounced pro-apoptotic activity and the ability to inhibit PIM-1 kinase intracellularly. These findings support the hypothesis that PIM-1 kinase represents a promising molecular target for the treatment of leukemia.

## 1. Introduction

Protein kinases are key mediators of signal transduction pathways, which are frequently dysregulated in cancer, making them among the most promising targets for drug development. Several chemical kinase inhibitors have shown potent clinical activity in tumors in which the target kinase is deregulated, which resulted in the approval of about 80 kinase inhibitors for clinical use in Europe [1]. The constitutively active serine/threonine kinases CK2 and PIM-1 show abnormally high expression levels in many cancers, particularly prostate, breast, and lung cancers, and in hematological cancers [2,3]. Studies conducted around the world have shown that the protein kinase CK2 as well as PIM-1 kinase represent new potential targets for anti-cancer therapies. Both CK2 and PIM kinases have been shown to play crucial roles in cancer transformation, and cancer cells are more sensitive than normal cells to the inhibition of their activity [4,5]; therefore, both kinases may represent interesting new molecular targets for cancer therapy [6]. Notably, numerous studies have identified PIM kinases as potential therapeutic targets in T-cell acute lymphoblastic leukemia (T-ALL) [7,8] as well as in human breast cancer [9]. Gene expression profiling using a publicly available T-ALL dataset revealed that PIM-1 is overexpressed in the majority of early T-cell precursor (ETP)-ALL cases as well as in a small subset of non-ETP ALL. Upregulation of PIM-1 mRNA and protein expression has been also observed in human breast cancer cells compared to normal breast tissue [9]. It has been reported that elevated PIM-1 protein levels were associated with increased malignancy and a higher tumor grade. These findings suggest that high PIM-1 expression may serve as a negative prognostic indicator, particularly in patients with recurrent or treatment-resistant breast cancers. In a recent 2024 study, Chen et al. highlighted the therapeutic potential of PIM kinases as anticancer targets [5]. While several PIM inhibitors have demonstrated promising results in preclinical studies, including SGI-1776, GDC-0339, CX-6258, TP-3654, AZD1208, and PIM447, these agents have yet to show significant progress in clinical development [5].

CK2 has been shown to play a key regulatory role in at least 15 cancer-related proteins, including the tumor suppressor p53, histone-modifying enzymes HDAC1 and HDAC2, and the NF-κB subunit RelA [10]. Upregulation of CK2 activity has been linked to tumor progression in multiple cancers, such as breast, lung, colon, and prostate, making it an attractive target in cancer therapy [11,12]. Importantly, CK2 has been implicated in the pathogenesis of various hematological cancers, including leukemia [13] as well as breast cancer [14] and its activity has been associated with poorer clinical outcomes. Numerous studies have shown that the oncogenic capacity of CK2 in some types of cancer is associated with the dysfunction of signaling pathways, such as the Wnt protein pathway, Hedgehog protein, the transcription factor NF-κB, JAK/STAT kinases, and PTEN/PI3K/Akt-PKB [15].

The involvement of CK2 in the progression of numerous diseases has made it a prominent therapeutic target. Over the past three decades, many potent, cell-permeable, ATP-competitive CK2 inhibitors have been developed [16]. Most advanced clinical inhibitors include Silmitasertib (CX-4945) which has been investigated clinically in combination with platinum-based therapy [17]. Another CK2-targeting agent in clinical trials is CIGB-300, which has also demonstrated encouraging therapeutic effects in hematological cancers as well as breast cancer [18].

The important group of ATP-competitive CK2 inhibitors are benzimidazole derivatives. Studies conducted by both our team and other teams show that compounds that are ATP-competitive CK2 inhibitors very often show similar activity against PIM-1 kinase, due to structural similarities within the pharmacophores of both kinases [19,20] and some of them act intracellularly as “dual inhibitors” [20]. 5,6-Dichloro-1-(β-D-ribofuranosyl)benzimidazole (DRB) described by Zandomeni et al. in 1986 [21] was the first structure among thi inhibitor family. Based on it, a number of derivatives showing effective CK2 kinase inhibition were developed, focusing on the removal of the sugar moiety from the molecule and the replacement of the chlorine with bromine. These include 4,5,6,7-tetrabromo-1*H*-benzimidazole (TBBi), 4,5,6,7-tetrabromo-1*H*-benzimidazole-2-*N*,*N*-dimethylamine (DMAT), 4,5,6,7-tetraiodo-1*H*-benzimidazole (TIBi), and 4,5,6,7-tetrabromo-2-methyl-1*H*-benzimidazole (2-Me-TBBi). Another group of CK2 inhibitors are 4,5,6,7-tetrabromo-1(2)*H*-benzotriazole (TBBt) derivatives. The interaction of TBBi and TBBt with the active site of CK2 involves the action of bromine atoms at positions 5 and 6 with residues Glu114 and Val116. In contrast, the nitrogen atom of imidiazole or triazole, respectively, forms a hydrogen bond with Lys68 or a water-mediated hydrogen bond with Glu81. TBBi has been shown to be more selective in differentiating molecular forms of CK2 compared to TBBt, suggesting its potential advantage as an inhibitor [22].

Considering that the ability of anti-cancer drugs to induce apoptosis is the most important feature, TBBi derivatives among polybrominated benzoazole derivatives are the most promising candidates in anti-cancer therapy. The structures of the most promising TBBi derivatives are shown in Figure 1. 2-Dimethylamino-4,5,6,7-tetrabromo-1*H*-benzimidazole (DMAT), described in 2004 [23], displays a very low Ki value for a CK2 inhibitor (40 nM); it is cell permeable and induces endogenous CK2 inhibition, which is several fold higher than that of TBBt. It has been shown that DMAT triggers apoptosis of Jurkat cells more readily than TBBt (DC_50_ value 2.7 vs. 17 µM), and unlike TBBt, it does not display any side effect on mitochondria polarization up to a 10 µM concentration [24]. Other studies have demonstrated that DMAT, in contrast to TBBt, induced reactive oxygen species (ROS) and DNA-double-strand-breaks (DSBs) that might contribute to the more effective induction of apoptosis [25]. Based on 4,5,6,7-tetrabromobenzotriazole (TBBt) and 2-dimethylamino-4,5,6,7-tetrabromo-1*H*-benzimidazole (DMAT) scaffolds for CK2 inhibition and a hydroxamate to coordinate the zinc atom present in the active site of HDAC (zinc binding group, ZBG), new multitarget inhibitors have been described [26]. Cellular assays on different cancer cell lines rendered promising results for *N*-hydroxy-8-(4,5,6,7-tetrabromo-2-(dimethylamino)-1*H*-benzo[d]imidazol-1-yl)octanamide (**A**). This inhibitor demonstrated the highest cytotoxic activity and pro-apoptotic effects against Jurkat cells, along with superior mitochondrial targeting and the ability to overcome multidrug resistance.

Among amino alcohol derivatives of parental DMAT described in Wińska et al., 2023 [27], 1,1,1-trifluoro-3-[(4,5,6,7-tetrabromo-1*H*-benzimidazol-2-yl)amino]propan-2-ol (**B**) exhibits the highest selectivity and most pronounced pro-apoptotic activity against the tested cancer cell lines—particularly acute lymphoblastic leukemia—and also induces autophagy in K-562 cells.

Other research demonstrated that the activity of tetrahalogenated benzimidazoles, including TBBi, against the hormone-sensitive human prostate cancer cell line LNCaP, carrying an aminoalkylamino group at position 2, was better than parental compounds. The activity of 2-aminoalkylamino derivatives was notably higher (LD_50_ 4.75–9.37 µM) than that of TBBi and TIBI (LD_50_ ≈ 20 µM). Among the tested compounds, *N1*-(4,5,6,7-tetrabromo-1*H*-benzimidazol-2-yl)-ethane-1,2-diamine (**C**) showed the strongest induction of apoptosis at 10 µM and 20 µM with concentration-dependent PARP cleavage [28].

It has been shown that 1-(β-D-2′-deoxyribofuranosyl)-4,5,6,7-tetrabromo-1*H*-benzimidazole, called K164 (also termed TDB), a TBBi derivative with a deoxyribose moiety, induces cytotoxic efficacy by inducing apoptosis in human T lymphoblastoid CEM cells [20]. Additional studies have revealed that K164 shows diverse cytotoxicity and pro-apoptotic efficacy in other cell lines, including myeloid leukemia and androgen-responsive prostate cancer [29]. Apoptosis was most strongly induced in the LNCaP cell line, with 91.8% apoptotic cells observed after 48 h of exposure to 10 μM K164. Other results have demonstrated that K164 effectively induces apoptosis in breast cancer cell lines, especially in the MCF-7 and SK-BR-3 cells [30].

Among a series of polybrominated benzimiadazole derivatives previously described by us [31], 4,5,6,7-tetrabromo-2-methyl-1*H*-benzimidazole (2-Me-TBBi) demonstrated the best pro-apoptotic activity toward leukemia cells, with even twenty-fold better pro-apoptotic activity than parental compound (TBBi). We have also noted that the pro-apoptotic properties of TBBi substituted at the *N1* position with various cyanoalkyl groups (cyanomethyl, cyanoethyl, cyanopropyl, and cyanobutyl) are reduced with a longer alkyl chain. The best pro-apoptotic effect was exhibited by 4,5,6,7-tetrabromo-2-methyl-1*H*-benzimidazol-1-yl)acetonitrile (**D**). Furthermore, when **D** was used in combination with 5-fluorouracil (5-FU) the increase in the pro-apoptotic activity towards leukemia was observed [32]. In a separate study, we showed that among *N*-hydroxypropyl TBBi and 2-Me-TBBi derivatives, as well as their corresponding esters, (4,5,6,7-tetrabromo-2-methyl-1*H*-benzimidazol-1-yl)propyl hexanoate (**E**) most effectively induced apoptosis in breast cancer cell lines MCF-7 and MDA-MB-231 [33]. In our most recent study, we described 1-phenyl-2-(4,5,6,7-tetrabromo-1*H*-benzimidazol-1-yl)ethanone (**F**), which exhibits strong pro-apoptotic activity by inducing the mitochondrial apoptotic pathway in CCRF-CEM cells [34]. These results are consistent with data showing that compound **F** reduces intracellular levels of the CK2α protein and inhibits CK2-mediated phosphorylation of Ser529 in NF-κB p65.

As a continuation of our previous works, we have synthesized and evaluated the anti-cancer as well as pro-apoptotic properties of *N*-substituted TBBi derivatives, including 2-oxopropyl or 2-oxobutyl substituents and their hydroxyl analogues. Their structures correlate with compound **F** (Figure 1), which exhibits good pro-apoptotic properties. We decided to investigate how the replacement of the benzene ring with a small methyl substituent and a slight modification of the alkyl chain between the TBBi scaffold and the carbonyl group would affect the anti-cancer properties and affinity of the compounds for CK2 and PIM-1.

## 2. Results

### 2.1. Synthesis

The anti-cancer potential of two new 4,5,6,7-tetrabromo-1*H*-benzimidazole derivatives, **1** and **2**, and two previously described compounds, **3** and **4**, was studied (Figure 1). Ketone **1** was obtained by *N*-alkylation of 4,5,6,7-tetrabromo-1*H*-benzimidazole (TBBi) with chloroacetone. The reaction was performed in acetone in the presence of K_2_CO_3_. The progress of the reaction was monitored by TLC. After 24 h, the reaction was stopped by filtering of the inorganic salts and product was purified by recrystallization from methanol. The yield of the reaction reached 53%. The next step involves a reduction of carbonyl compound **1** to alcohol **2** with the use of NaBH_4_ in methanol at room temperature. The progress of the reaction was controlled by TLC. The 100% conversion of the substrate was achieved after 24 h. After recrystallization from methanol, alcohol **2** was obtained with a yield of 70%. Ketone **3** and alcohol **4** were obtained according to a method previously described by us [35]. 4-(4,5,6,7-Tetrabromo-1*H*-benzimidazol-1-yl)butan-2-one **3** was prepared by Michael addition of TBBi to methyl vinyl ketone (yield 66%) and then reduced to the appropriate alcohol **4** using NaBH_4_ as reducing agent and methanol as a solvent (yield of 86%).

### 2.2. Biological Evaluation

#### 2.2.1. Determination of the Inhibitory Activity of Compounds 1–4 Towards CK2 and PIM-1 Kinases

The in vitro inhibitory activity of the newly synthesized compounds and the parent compound (TBBi) against the recombinant human CK2 catalytic subunit (rhCK2α) and PIM-1 kinase was evaluated using a radiometric assay. Synthetic peptide RRRADDSDDDDD served as the substrate for CK2, while peptide ARKRRRHPSGPPTA was used as the substrate for PIM-1. None of the tested compounds showed better inhibitory potency against CK2α than the parental compound, TBBi (Table 1). The newly obtained compounds were more efficient inhibitors of PIM-1 kinase than CK2 kinase with the lowest IC_50_ i.e., 0.52 µM value obtained for derivative **1**. This derivative was also the most efficient inhibitor of CK2α among the newly obtained compounds with an IC_50_ equal to 6.65 µM.

#### 2.2.2. Cytotoxic Effect of TBBi Derivatives Toward Cancer Cell Lines, CCRF-CEM, MCF-7, and MDA-MB-231, and Non-Cancerous Vero Cells

The results show that among newly obtained compounds, only derivatives **1**, **2**, and **4** inhibit CK2/PIM-1 kinases, whereas cpd. **3** is inactive. The following studies concerning anti-cancer activity of the newly obtained TBBi derivatives were devoted to these three compounds without cpd. **3**. The cytotoxicity of the newly obtained inhibitors, i.e., **1**, **2**, and **4** was tested towards four cell lines, i.e., CCRF-CEM (acute lymphoblastic leukemia ALL), MCF-7 (breast carcinoma), MDA-MB-231 (triple negative breast carcinoma), and Vero (*Cercopithecus aethiops* kidney). The representative plots demonstrating sigmoidal doseresponse curves for the derivatives **1**, **2**, and **4** and the parental compound, TBBi, are shown in Figure 2. The IC_50_ values, describing the half maximal effective concentration of each tested compound, were calculated and summarized in Table 2. The results obtained for TBBi describing cellular viability of the tested cells are comparable with those obtained so far [33,34].

Based on IC_50_ values, we concluded that only one derivative demonstrated better cytotoxicity than TBBi towards CCRF-CEM cells, i.e., compound **1** with an IC_50_ value of 12.43 µM. Similarly, among newly synthetized derivatives, compound **1** was the most active against MCF-7 and MDA-MB-231 with IC_50_ values equal to 17.09 µM and 21.20 µM. However, for MCF-7, none of the newly obtained derivatives was better than the parental compound, as IC_50_ values for **1**, **2**, and **4** were in the range from 17.09 µM to 41.86 µM, whereas the IC_50_ for TBBi was 12.61 µM. The lowest and highest IC_50_ values for non-cancerous Vero cells were obtained for compounds **1** (20.20 µM) and **4** (132 µM), respectively. Among all the studied cancer cell lines, MDA-MB-231 cells indicated the highest resistant to the tested compounds. Additionally to IC_50_ values, the selectivity indexes (SI) were calculated for the tested compounds (Table 2). Interestingly, all newly obtained derivatives demonstrated better selectivity towards tested cancer cells than TBBi, with the highest SI value equal to 5.21 obtained for derivative **2** in MCF-7 cells. Considering that among newly obtained derivatives, compound **1** was the best inhibitor of CK2α and PIM-1 kinase, it effectively affected the viability of both CCRF-CEM and MCF-7 cell lines. The following biological studies are devoted to effect of derivative **1** and TBBi (the parental compound) on CCRF-CEM as well as MCF-7.

#### 2.2.3. Induction of Apoptosis in CCRF-CEM and MCF-7 Cells

To assess the pro-apoptotic effects of compound **1** and TBBi, we analyzed annexin V binding to phosphatidylserine by flow cytometry. The tested compounds were used in three concentrations corresponding to their 0.5 IC_50_, IC_50_, and 2 IC_50_ values, i.e.,: 8 µM, 16 µM, and 32 µM. The results obtained for CCRF-CEM and MCF-7 are shown in Figure 3 and Figure 4, respectively. The obtained results demonstrated dose-dependent induction of apoptosis in all treated cells with the highest proportion of apoptotic CCRF-CEM cells, close to 98% after 32 µM **1** (Figure 3b) and 95% after 32 µM TBBi. After 16 µM TBBi treatment, 44% of leukemic cells were apoptotic, whereas after 16 µM cpd. **1**, 79.6% of cells were apoptotic. Furthermore, treatment of CCRF-CEM cells with 8 µM of compound **1** induced apoptosis in 23.5% of cells (a statistically significant effect), while 8 µM of TBBi induced apoptosis in 16% of cells (not statistically significant). The results demonstrated that the newly synthesized TBBi derivative, compound **1**, exhibited stronger pro-apoptotic activity against CCRF-CEM cells compared to the parent compound, TBBi.

Similarly to leukemia cells, compound **1** induced apoptosis in MCF-7 cells more effectively than the parental compound, TBBi, at all tested concentrations, and the percent of apoptotic cells was in the range from 28.6% to 54.29%, whereas after TBBi treatment, the range of apoptotic cells was 16.5% to 47% (Figure 4).

#### 2.2.4. Mitochondrial Membrane Potential (ΔΨm) in CCRF-CEM and MCF-7

Given the significant pro-apoptotic effect of compound **1** on CCRF-CEM and MCF-7 cells, we next measured the mitochondrial membrane potential (ΔΨm) in both cell lines after 48 h of treatment with compound **1** and TBBi (Figure 5). Incubation with both compounds induced mitochondrial membrane depolarization in a dose-dependent manner, as indicated by a shift from red to green fluorescence in CCRF-CEM cells and in MCF-7 cells treated with TBBi. In MCF-7 cells, treatment with compound **1** resulted in 26% of cells showing reduced ΔΨm at 16 µM and 19% at 32 µM. The most pronounced effect was observed in CCRF-CEM cells treated with 32 µM of compound **1**, where 99.4% of cells exhibited reduced mitochondrial membrane potential (ΔΨm). Additionally, treatment with 16 µM of compound **1** decreased ΔΨm in 76% of CCRF-CEM cells, compared to only 37% with the same concentration of TBBi. These results align well with the annexin V binding studies of compound **1**-treated CCRF-CEM cells and support the involvement of the intrinsic apoptotic pathway.

#### 2.2.5. The Effect of Cpd. **1** and TBBi on Cell Cycle Progression in CCRF-CEM and MCF-7 Cells

Since CK2 and PIM kinases regulate cell cycle progression, their inhibition may alter the distribution of cells across different cell cycle phases. Therefore, we examined cell cycle progression in CCRF-CEM and MCF-7 cells following treatment with compound **1** and TBBi. Representative plots showing the percentage of cells in each phase are presented in Figure 6. The cell cycle changes observed in CCRF-CEM cells indicate the shortening of both S and G2/M phases in all treated cells with the most significant effect, i.e., 20% and 19% cells in the S phase observed after treatment with 16 µM **1** and TBBi, respectively, whereas 33% cells were in the S phase in the control. For the G2/M phase, the most significant result, i.e., 10% cells in the G2/M phase, was observed in cells treated with 16 µM of **1**. For control cells, the percentage of G2/M cells was 17%. In addition to those changes, we observed cells in the sub-G1 phase, which are apoptotic cells. Accumulation of cells in this phase reflects DNA degradation characteristic of apoptotic cells. The largest percentage of these cells, i.e., 32% was determined in 16 µM 1-treated cells (Figure 6a,b). This result confirms the Annexin V binding results, indicating better pro-apoptotic properties of cpd. **1** than TBBi.

Similarly to CCRF-CEM cell line, we observed a statistically significant decrease in MCF-7 cells in the S phase after treatment with both tested compounds (Figure 6c,d), with the strongest effect, i.e., 11% and 12% cells in the S phase after treatment with 8 µM **1** and TBBi, respectively, whereas the percentage of S phase cells in the control was 17%. Additionally, we observed the prolongation of G2/M in MCF-7 treated with **1** at the highest concentration. The obtained results indicate that cpd. **1** induced G2/M phase arrest in MCF-7 with the highest effect detected for the 16 µM concentration, manifested as 32% cells in that stage, whereas in the control, the percentage of G2/M cells was 22%.

#### 2.2.6. Activation of Caspase-3 in CCRF-CEM

Since caspase-3 is a key protease activated in the early stages of apoptosis, we next investigated whether apoptosis in CCRF-CEM cells is mediated through caspase-3 activation. Both cpd. **1** and TBBi induce caspase-3 activation in a dose-dependent manner in CCRF-CEM (Figure 7); however, the most significant effect, i.e., 60% cells with the active caspase, was observed for CCRF-CEM treated with 16 µM cpd. **1**. The results are consistent with the previously observed pro-apoptotic properties of derivative **1**.

#### 2.2.7. Intracellular Inhibition of CK2 and PIM-1 Kinases in CCRF-CEM and MCF-7 Cells

Based on the observed inhibition of recombinant CK2 and PIM-1 kinases by derivative **1**, along with its pro-apoptotic effects, we evaluated the impact of compound **1** and TBBi on these kinases in CCRF-CEM and MCF-7 cells. After 24 h of treatment, we assessed site-specific phosphorylation of Ser529 in NF-κB p65 and Ser112 in BAD—a marker of PIM kinase activity—using Western blot analysis (Figure 8). The data for BAD confirmed intracellular inhibition of this kinase by the tested compounds in both cell lines in a dose-dependent manner. A statistically significant decrease in phosphorylated BAD was observed following treatment with 16 µM of compound **1**, with relative protein levels reduced to 0.48 and 0.46 in CCRF-CEM and MCF-7 cells, respectively (Figure 8a–c). TBBi acted as a weaker inhibitor of PIM in both lines, with relative levels of 0.8 and 0.7 in CCRF-CEM and MCF-7, respectively. Interestingly, inhibition of CK2 was more effective in cells treated with cpd. **1** than TBBi, with the best effect determined as the relative level of phosphorylated p65 observed in CCRF-CEM treated with 16 µM concentration with a value of 0.46. The weakest CK2 inhibitory effect was observed in MCF-7 cells with the lowest relative level of p65-P determined after 8 µM **1** with a value 0.84.

To confirm the significant reduction in NF-κB p65 Ser529 phosphorylation observed in CCRF-CEM cells, we employed the PhosFlow method to measure the percentage of cells with unphosphorylated Ser529 in NF-κB p65 after 24 h of treatment with compound **1** and TBBi (Figure 9). The significant inhibition of CK2 with 29% and 28% of p65-P-negative cells was observed after treatment with 16 µM **1** and TBBi, respectively.

### 2.3. Molecular Modeling

In order to gain molecular insight into the results of in vitro enzymatic assays, comprehensive in silico docking calculations were performed, followed by molecular dynamics simulations. The docking results for compounds **1**, **2**, and **4** and TBBi are shown in Appendix A. Molecular dynamics results, including C***α*** RMSD and ligand RMSD fluctuations, are shown in Appendix A. Interaction analysis of compound **1** docked to PIM-1 showed 2 hydrogen bonds (with ASP186 and LYS67), in contrast to one hydrogen bond (with LYS67) established by TBBi, while compounds **2** and **4** had no hydrogen bonds. Compounds **1**, **2**, and TBBi had one Pi-Sigma interaction. All ligands had substantial van der Waals and alkyl interactions as well. Based on these results, compound **1** renders more promising than **2** and **4** for PIM-1 inhibition.

A similar interaction analysis was performed for CK2α, showing that only compounds **1** and **2** established a single hydrogen bond (with ASN118). Compounds **1** and **4** had two Pi-Sigma interactions, while compound **2** and TBBi only had one. Additionally compound **1** established one halogen interaction with Br. Similarly to PIM-1, all ligands also had many van der Waals and alkyl interactions with CK2α. These results suggest compound **1** as the most promising for CK2α inhibition.

For each complex, one representative frame was selected from the most stable 20ns simulation window from the MD trajectories (Appendix A). Structural comparison of the TBBi best docking pose and the MD result for both PIM-1 and CK2α is shown in Figure 10. Structural comparison of compound **1** and TBBi MD-derived poses, as well as 2D protein–ligand interaction mapping diagrams are presented in Figure 11 and Figure 12. Protein–ligand interaction analysis for compound **1** showed that compound 1 established one hydrogen bond for both PIM-1 (with ASP154) and CK2α (with ASN117) kinases. For compound **1** docking to PIM-1, there were 6 alkyl interactions and 10 van der Waals contacts. For compound **1** docking to CK2α, there were 5 alkyl interactions and 6 van der Waals contacts; moreover, there was a carbon hydrogen bond possible (with LEU44). TBBi also created one hydrogen bond (with GLU89) when docked to PIM-1. TBBi docked to CK2α did not establish hydrogen bonds but had a Pi-Sigma interaction (with VAL52). These findings are in agreement with obtained experimental results, suggesting hydrogen bond formation as the dominant force discriminating the affinity of compounds to respected targets.

## 3. Discussion

Four *N*-substituted TBBi derivatives, including 2-oxopropyl/2-oxobutyl substituents or their appropriate hydroxyl analogues, were synthesized, and their anti-cancer properties were evaluated in this work. All compounds were obtained with satisfactory yields, ranging from 66% to 86%, and in a relatively short time. Although the results of in vitro CK2/PIM-1 inhibition studies revealed that compounds 1, 2, and 4 decreased the catalytic activity of both recombinant enzymes to a lower or comparable extent as a parental TBBi, the selectivity of the tested compounds against CCRF-CEM as well as MCF-7 cells lines were even better than for the parental cpd. TBBi. The cytotoxicity of the newly obtained compounds correspond to their ability to inhibit of CK2/PIM, as the most cytotoxic derivative, i.e., 1 demonstrated the lowest IC_50_ for both CK2 and PIM-1 kinases, whereas less active derivatives against recombinant kinases cpd. **2** and cpd. **4** were generally less cytotoxic. Moreover, the molecular docking analysis paired with MD calculations rationalized the experimental data by demonstrating the putative molecular mechanisms that render compound **1** a strong candidate for PIM-1 inhibition and a weak binder for CK2α. The in silico results for compound **1** docking to PIM-1 and CK2α, followed by molecular dynamics simulations, suggests better stabilization of the compound **1**-PIM-1 complex and thus more stable binding. These findings confirmed that efficient inhibitors of recombinant PIM-1 kinase should possess the ability to form stable hydrogen bonds and maintain favorable hydrophobic interactions within the binding pocket. Compound **1** demonstrates a consistent interaction profile, with its hydrogen bond to ASP154 in PIM-1 and ASN117 in CK2α remaining stable throughout the simulation window. This, along with compound **1** extensive van der Waals and alkyl interactions, supports hints towards binding stability, particularly toward PIM-1. The molecular dynamics simulations successfully corrected the initial docking pose of TBBi in CK2α, rotating it into a conformation that closely aligns with the experimentally determined crystal structure [36] and provide informative molecular insight into experimentally obtained inhibitory results.

Among the tested tumor cell lines, MDA-MB-231 was the most resistant to the tested compounds with IC_50_ values even higher that those obtained for normal Vero cells. That result is in good agreement with our previous data and also the literature data demonstrating the drug resistance of this cell line [37]. Regarding the cytotoxicity against tumor cells as well as the ability to inhibition of recombinant CK2/PIM-1 kinases, further investigations of biological action of the most promising compound, i.e., cpd. **1**, were performed. Considering that CK2 as well as PIM-1 kinase are anti-apoptotic and promote cell survival and proliferation, we focused on the study of apoptosis induction in CCRF-CEM and MCF-7 by cpd. **1**. Among the tested cell lines, derivative 1 induced the highest level of apoptosis in CCRF-CEM cells. Its pronounced pro-apoptotic activity correlated with a strong reduction in mitochondrial membrane potential (ΔΨm), indicating activation of the intrinsic apoptotic pathway [38]. The obtained results concerning caspase-3 activation strongly correlate with the pro-apoptotic properties of cpd. **1**. Caspase-3 is a key executioner caspase that becomes activated in the early stages of apoptosis and proteolytically cleaves and activates other caspases, as well as crucial cytoplasmic (e.g., D4-GDI, Bcl-2) and nuclear (e.g., PARP) targets [39]. Additionally, the cell cycle progression results demonstrate that cpd. **1**-treated CCRF-CEM cells in the sub-G1 phase correspond to cells with reduced DNA content and morphologic changes, including nuclear condensation [40]. These results are in good agreement with our previous observations showing that the pro-apoptotic efficacy of TBBi derivatives is always better towards leukemia cells than MCF-7 cells. The differing susceptibility to apoptosis may be attributed to caspase-3 deficiency in MCF-7 cells, which makes them more resistant to apoptosis induction, particularly through the caspase-3-dependent pathway [27,41].

Considering that cpd. **1** demonstrated activity against recombinant PIM-1 kinase similar to the parental compound and an even weaker ability to inhibit CK2 kinase, we hypothesized that its pro-apoptotic activity could be connected to the better intracellular activity of this derivative than TBBi. To verify this hypothesis, we studied the intracellular activity of CK2 as well as PIM-1 kinases in CCRF-CEM and MCF-7 cells. The obtained results verified that cpd. **1** inhibited both tested kinases in the studied cell lines to a higher extent than the parental compound. The immunodetection results are consistent with literature reports showing that both CK2 and PIM kinases protect cells from apoptosis by phosphorylating a broad range of proteins involved in the apoptotic response [5,42]. It has been shown that PIM-1-mediated phosphorylation of BAD at Ser112 inactivates the protein, thereby enhancing the anti-apoptotic function of BCL-2 and promoting cell survival [43]. Phosphorylation of BAD promotes its binding to 14-3-3 proteins, thereby preventing its interaction with the anti-apoptotic proteins Bcl-2 and Bcl-xL [44]. The results support the hypothesis that PIM-1 plays a critical role in regulating survival signaling by inhibiting mitochondrial pro-apoptotic Bcl-2 family members, such as BAD [45]. Moreover, our results demonstrating activation of caspase-3 after cpd. **1** treatment in CCRF-CEM correspond well with other studies showing that the tumorigenic activity of PIM-1 synergizes with c-MYC in lymphomagenesis through the phosphorylation and inactivation of the pro-apoptotic protein BAD, thereby preventing the activation of the key executioner caspase, caspase-3 [46].

Cell cycle progression studies indicate the similar effect of cpd. **1** on the shortening of the S phase in both tested cell lines and the cell line-dependent effect on the G2/M phases, i.e., shortening in CCRF-CEM cells and prolongation in MCF-7 cells, respectively. The shortening of the S phase upon TBBi derivative treatment was observed by us in MCF-7, earlier [34]. However, G2/M arrest occurred in MCF-7 after cpd. **1** treatment, which is in contrast to our previous studies showing G1 arrest in MCF-7 after inhibition of CK2-mediated phosphorylation of NF-κBp65 by TBBi derivatives [27,34]. Moreover, the literature data have shown that treatment of cells with a selective inhibitor of CK2, i.e., CX-4945 5-(3-chlorophenylamino)benzo[c][2,6]naphthy-ridine-8-carboxylic acid) [47] results in reduced phosphorylation of a key cell-cycle inhibitor protein [p21 (T145)] and increases the stability and levels of total p21 and p27, cell cycle inhibitors responsible for stopping the cell cycle in the G1 phase, activation of repair mechanisms, and apoptosis of cells with damaged DNA [48]. Interestingly, the similar effect of the decreased G2/M phase was observed in K562 cells (chronic myeloid leukemia) after treatment with SMI-4a, a selective PIM-1 kinase inhibitor that inhibits PIM-1 kinase activity in vivo and in vitro [49]. The difference between previously described inhibitors and cpd. **1** in the effect on cell cycle progression can be connected with the inhibition of PIM-1 kinase, as so far, none of the TBBi derivatives tested by us have not decreased phosphorylation of Ser 112 in BAD. It has been shown that estradiol-induced PIM-1 overexpression in breast cancer leads to the phosphorylation of key proteins, which in turn suppresses the expression of cell cycle inhibitors (CDKN1A and CDKN2B) [50]. PIM kinases phosphorylate CDK inhibitor p27 at two specific sites, namely Thr157 and Thr198. This phosphorylation event enhances cell cycle progression and subsequent proteasomal destruction [51].

## 4. Materials and Methods

### 4.1. Materials

^1^H (600 MHz) and ^13^C NMR (150 MHz) spectra were recorded on a JEOL JNM-ECZL in DMSO-d_6_ solution; chemical shifts (δ) are reported in ppm; coupling constants (*J*) are given in hertz (Hz). The reactions were monitored by thin-layer chromatography (TLC) aluminum plates with silica gel Kieselgel 60 F_254_ (0.2 mm thickness film, Merck, Darmstadt, Germany) using UV light as a visualizing agent. HRMS spectra were recorded on a Micro-mass ESI Q-ToF Premier instrument (Micromass UK Limited, Manchester, UK. Melting points were determined in open glass capillary tubes and are uncorrected. All reagents, solvents and chemicals were purchased from Avantor (Radnor, Pennsylvania, USA), Merck (Darmstadt, Germany) and Sigma-Aldrich (St. Louis, MO, USA) and were used at analytical grade.

### 4.2. Chemical Study

#### 4.2.1. Synthesis of 1-(4,5,6,7-Tetrabromo-1*H*-benzimidazol-1-yl)propan-2-one (**1**)

A solution of 4,5,6,7-tetrabromo-1*H*-benzimidazole (1.15 mmol, 0.5 g), chloroacetone (1.85 mmol, 0.15 mL), and K_2_CO_3_ (1.85 mmol, 0.25 g) in acetone (13 mL) was stirred at reflux. The progress of the reaction was monitored by TLC using chloroform:methanol (9:1 *v*/*v*) as the eluent. After 24 h, the reaction mixture was cooled to room temperature, and the inorganic solid was filtered off and washed with acetone (2 × 10 mL). Then, the organic solvent was evaporated under reduced pressure. The product was dissolved in CH_2_Cl_2_ and washed with water (3 × 10 mL). The organic phase was dried under MgSO_4_. After filtering off the MgSO_4_ and evaporating the solvent, the obtained solid was recrystallized from methanol.

1-(4,5,6,7-tetrabromo-1*H*-benzimidazol-1-yl)propan-2-one (**1**). Yield 53%, colorless crystals mp 210–212 °C. ^1^H NMR (DMSO-d_6_) σ ppm: 2.30 (s, 3H, CH_3_), 5.51 (s, 2H, CH_2_), 8.32 (s, 1H, CH). ^13^C NMR (DMSO-d_6_) σ ppm: 26.97, 56.50, 107.29, 117.13, 121.06, 122.89, 132.32, 143.75, 149.47, 202.49. HRMS [M+H^+^] *m/z* calcd for C_10_H_7_Br_4_N_2_O + 490.7245, found 490.7245.

#### 4.2.2. Synthesis of 1-(4,5,6,7-Tetrabromo-1*H*-benzimidazol-1-yl)propan-2-ol (**2**)

To the cooled and stirred mixture of appropriate ketone 1 (0.31 mmol, 0.15 g) in methanol (1.5 mL), sodium borohydride (0.62 mmol, 0.024 g) was added. Then, the reaction mixture was stirred at room temperature for 24 h. The progress of the reaction was monitored by TLC, using chloroform: methanol (9:1 *v*/*v*) as the eluent. After 24 h the solvent was evaporated and 20 mL of H_2_O was added. The alcohol was extracted with CH_2_Cl_2_ (5 × 30 mL), and the organic layer was washed with water (3 × 30 mL), dried over anhydrous MgSO_4_, and evaporated. The product was recrystallized from methanol.

1-(4,5,6,7-tetrabromo-1*H*-benzimidazol-1-yl)propan-2-ol (**2**). Yield 70%, colorless crystals mp 237–239 °C. ^1^H NMR (DMSO-d_6_) σ ppm: 1.14 (d, 3H, CH_3_, *J* = 6 Hz), 3.96 (brs, 1H, OH), 4.12–4.16 (m, 1H, CH), 4.62–4.65 (m, 1H, CH_2_), 4.97–4.98 (m, 1H, CH_2_), 8.35 (s, 1H, NH). ^13^C NMR (DMSO-d_6_) σ ppm: 20.37, 53.34, 66.80, 106.55, 116.45, 120.15, 122.18, 131.48, 143.71, 149.61. HRMS [M+H^+^] *m/z* calcd for C_10_H_9_Br_4_N_2_O + 492.7402, found 492.7401.

### 4.3. Biological Assays

#### 4.3.1. Reagents and Antibodies

Dimethyl sulfoxide (DMSO; molecular biology grade), used as a solvent for all stock solutions of chemical agents, was obtained from Sigma-Aldrich (St. Louis, MO, USA). Reagents for flow cytometry were purchased from BD Biosciences Pharmingen (San Diego, CA, USA). The primary antibodies used included: anti-GAPDH (Sigma-Aldrich; #MAB374, 1:1000, overnight at 4 °C), anti-p65-PSer529 (Biorbyt, Cambridge, UK; #orb14916, 1:500, overnight at 4 °C), anti-NF-κB p65 (Biorbyt; #orb214507, 1:1000, overnight at 4 °C), anti-BAD-PSer112 (Abcam, Cambridge, UK; #ab129192, 1:500, overnight at 4 °C), and anti-BAD (Abcam; #ab62465, 1:1000, overnight at 4 °C). The secondary antibodies used were: Goat Anti-Rabbit IgG HRP-conjugate (Merck Millipore, Billerica, MA, USA; #12-348, 1:2000, 1 h at room temperature) and Goat Anti-Mouse IgG HRP-conjugate (Merck; #12-349, 1:2000, 1 h at room temperature). Halt™ Protease Inhibitor Cocktail (100×), (#78429) was obtained from ThermoFisher Scientific (Waltham, MA, USA). Additional reagents included NF-κB p65 (pS529)-PE (#558423), BD PhosFlow™ Perm Buffer III (#558050), Cytofix Fixation Buffer (#554655), and Stain Buffer (FBS; #554656), all from BD Biosciences Pharmingen (San Diego, CA, USA). Nitrocellulose membranes were obtained from GE Healthcare Life Sciences (Freiburg, Germany), and chemiluminescent HRP substrates from Bio-Rad (Hercules, CA, USA). Other chemicals and solvents were sourced from POCH (Avantor Performance Materials, Gliwice, Poland), Merck, and Sigma-Aldrich.

#### 4.3.2. Cloning, Expression, and Purification of Human CK2α, holoCK2, and PIM-1

CK2α, holoCK2, and PIM-1 were obtained according to Borowiecki [43] and Chojnacki [27]. The final protein concentrations were 12.68 mg/mL for CK2α, 1.61 mg/mL for holoCK2, and 3.00 mg/mL for PIM-1, as determined by the Bradford assay using bovine serum albumin as a standard [52].

#### 4.3.3. Inhibition of Recombinant CK2 and PIM-1

The synthesized compounds were evaluated for their inhibitory activity against human CK2α, the human CK2 holoenzyme, and PIM-1 using the P81 filter-based isotopic assay, as previously described [27]. IC_50_ values for the tested compounds were determined across eight concentrations ranging from 0.005 to 400 µM. The experimental data were fitted to a dose–response curve using non-linear regression analysis (variable slope) Y = Bottom + (Top − Bottom)/(1 + 10^((LogIC50 − X) × HillSlope^) equation in GraphPad Prism (Prism 9, version 9.0.1).

#### 4.3.4. Cell Culture and Agents’ Treatment

The CCRF-CEM (acute lymphoblastic leukemia, ALL) and MDA-MB-231 (triple-negative breast adenocarcinoma) cell lines were obtained from the European Collection of Authenticated Cell Cultures (ECACC), while the MCF-7 (hormone-dependent breast adenocarcinoma) and Vero (*Cercopithecus aethiops* kidney) cell lines were purchased from the American Type Culture Collection (ATCC, Manassas, VA, USA). CCRF-CEM cells were maintained in RPMI-1640 medium, whereas MCF-7 and MDA-MB-231 cells were cultured in DMEM (Sigma-Aldrich Chemical Company, St. Louis, MO, USA). Vero cells were cultured in minimum essential medium eagle (MEM; Sigma-Aldrich). All media were supplemented with 10% fetal bovine serum (FBS), 2 mM L-glutamine, and antibiotics (100 U/mL penicillin and 100 µg/mL streptomycin) (Sigma-Aldrich). Cells were grown in 75-cm^2^ culture flasks (Googlab Scientific, Rakocin, Poland) in a humidified atmosphere containing 5% CO_2_ at 37 °C. All experiments were performed using exponentially growing cultures. Stock solutions of the test compounds were prepared in DMSO and stored at –80 °C for up to one month. For cytotoxicity assays, stock solutions were diluted 200-fold in the appropriate culture medium to achieve the desired final concentrations, ensuring that the final DMSO concentration did not exceed 0.5%.

#### 4.3.5. 3-(4,5-Dimethylthiazol-2-yl)-2,5-diphenyltetrazolium Bromide (MTT)-Based Viability Assay

Following incubation with the tested compounds, the MTT assay was carried out as previously described [27]. Optical density was measured at 570 nm using a BioTek microplate reader (Winooski, VT, USA). All measurements were performed in at least three independent biological replicates.

#### 4.3.6. Detection of Apoptosis by Annexin V/Propidium Iodide (PI) Labelling

CCRF-CEM cells were seeded in 24-well plates at a density of 2 × 10^5^ cells/mL. The cells were treated with the tested compounds at concentrations of 8, 16, and 32 µM, followed by incubation for 48 h. After treatment, the cells were harvested and processed according to the previously described protocol [27]. Flow cytometric measurements were conducted within 1 h of cell labeling. Viable, necrotic, early apoptotic, and late apoptotic cells were identified using a BD Accuri C6 Plus flow cytometer and analyzed with BD Accuri C6 Plus software, version 1.0.34.1 (BD Biosciences, San Jose, CA, USA).

#### 4.3.7. Mitochondrial Membrane Potential (ΔΨm) Assay

The mitochondrial membrane potential was evaluated by flow cytometry using 5,5′,6,6′-tetrachloro-1,1′,3,3′-tetraethylbenzimidazolocarbocyanine iodide (JC-1; Sigma-Aldrich), as previously described [34].

#### 4.3.8. Caspase-3 Activation

Caspase-3 activity was measured using the FITC Active Caspase-3 Apoptosis Kit (BD Pharmingen, San Jose, CA, USA) according to the manufacturer’s instructions. The stained cells were measured with a BD Accuri C6 Plus flow cytometer.

#### 4.3.9. Detection of Cell Cycle Progression by Flow Cytometry

CCRF-CEM and MCF-7 cells were cultured in 6-well plates and treated with the tested compounds for 48 h. After exposure to the compounds, the cells were collected, washed with cold PBS, and fixed in 70% ethanol at –20 °C for at least 24 h. Subsequently, the cells were washed in PBS and stained according to the procedure previously described by our group [28]. The cellular DNA content was measured by flow cytometry using a BD Accuri C6 Plus flow cytometer (BD Biosciences, San Jose, CA, USA). The resulting DNA histograms were analyzed with BD Accuri C6 Plus software (San Diego, CA, USA) to assess the distribution of cells across different phases of the cell cycle.

#### 4.3.10. Western Blotting

Exponentially growing CCRF-CEM or MCF-7 cells were seeded at a density of 6 × 10^5^ cells per 6 cm dish. Compounds were then added, with a final DMSO concentration of 0.5%. Cell lysates were prepared as previously described [28]. The ECL substrate was used for signal detection, and the immunoblots were scanned using a G:BOX Chemi imaging system (Syngene, Cambridge, UK) equipped with GeneSys software, Version 1.4.3.0 (Syngene, Cambridge, UK).

#### 4.3.11. Densitometry

For densitometric analysis, immunoblots were scanned using the G:BOX Chemi system (Syngene), and the band intensities of phosphorylated and total proteins were quantified with GeneSys software (Syngene, Cambridge, UK). The levels of phosphorylated proteins were normalized to GAPDH, with the value for untreated cells set to 1, and subsequently expressed as a percentage relative to the appropriate control.

#### 4.3.12. PhosFlow Analysis

CCRF-CEM cells were prepared following the BD PhosFlow™ Protocol for Human PBMCs using NF-κB p65 (pS529)-PE (#558423), BD PhosFlow™ Perm Buffer III (#558050), Cytofix Fixation Buffer (#554655), and Stain Buffer FBS (#554656), all obtained from Becton Dickinson. Flow cytometric analysis was performed using a BD Accuri C6 Plus cytometer (Becton Dickinson) according to the manufacturer’s instructions.

#### 4.3.13. Statistical Evaluation

The results are presented as the mean ± standard error of the mean (s.e.m.) from at least three independent experiments. Statistical analysis was conducted using GraphPad Prism 5.0 software (GraphPad Software Inc., San Diego, CA, USA), with significance assessed by one-way ANOVA. The statistical significance of differences were indicated in figures by asterisks as follows: * *p* ≤ 0.05, ** *p* ≤ 0.01, and *** *p* ≤ 0.001.

### 4.4. In Silico Studies

#### 4.4.1. Molecular Docking

Compounds **1**, **2**, **4** and TBBi were docked to human protein kinases crystal structures PIM-1 (PDB code: 4DTK) and CK2α (PDB code: 4KWP) in AutoDock Vina v. 1.1.2 [53]. Compound **3** was not included in docking simulations because it is a very weak CK2α and PIM-1 inhibitor based on the experimental results. Ligands in non-ionizable form were prepared using Avogadro 1.2.5 [54]. Subsequently, the Gasteiger partial charges were calculated with AutoDock Tools v. 1.5.7 [55] and the flexibility of each ligand’s torsional angles, rotatable bonds and non-polar hydrogens were determined. Protein structures were preprocessed with ChimeraX v. 1.8 [56] (solvent, ions, and other ligands removed) and PDBFixer v. 1.11 [57].

For docking **1**, **2**, **4**, and TBBi to the PIM-1 target, a grid cubic box of size 10A and positioned on the center of mass of the crystal structure ligand was used. Similarly docking of **1** and TBBi to CK2α was performed. The resulting docking poses were clustered and ranked according to the AutoDock Vina score and root mean square deviation (RMSD) of the best pose (Appendix A).

#### 4.4.2. Molecular Dynamics

Based on the experimental results and interaction assessment of docking poses, compound **1** was selected as the best candidate for PIM-1 and CK2α inhibition. Further molecular dynamics (MD) analyses were conducted with compound **1** and TBBi as a reference compound. For both targets, the pose with the lowest Vina score was chosen as input orientations of compound **1** and TBBi for MD simulations. Preprocessing and simulation setup was done with OpenMMDL v. 0.9.2.4 [58]. For each complex, three independent 100 ns MD simulations were performed with OpenMM v. 8.2.0 [59] using the AMBER ff14SB [60] forcefield and TIP3P [61] water model. NaCl ions were added to 0.15 M bulk ionic strength, Hydrogens for pH = 7.0, and a cubic water box was positioned 1 nm around the complex. After system minimization, NVT and NPT equilibration was done. Production simulation was performed in NPT ensemble in 300 K and 1 atm with 0.002 ps step size and 1.0 ps-1 friction coefficient. Electrostatic interactions were treated using the PME method with a 1 nm cutoff distance and bonds involving hydrogen constraints.

The production of MD trajectories were preprocessed by treating PBC conditions and aligning the frames, choosing the first frame as a reference in MDTraj [62] and MDAnalysis [63]. For protein–ligand interaction analysis, the initial 20 ns of each trajectory was excluded from this analysis to allow for further system equilibration. The next 20 ns window with a stable RMSD was selected using the sliding windows approach with a 1 ns step. For protein–ligand interaction analysis, protein Cα and ligand RMSD changes in the simulation were calculated. Two-dimensional ligand interaction visualizations were generated in BIOIVIA Discovery Studio Visualizer 2025 (Dassault Systèmes) [64].

## 5. Conclusions

Based on the experimental results, compound **1** emerges as the most potent inhibitor of both PIM-1 and CK2α kinases when compared to compounds **2**, **3**, and **4** and is also a slightly more effective inhibitor of PIM-1 than the parent TBBi compound. The insertion of an additional CH_2_ group between the benzimidazole scaffold and carbonyl group (i.e., compound **1** vs. compound **3**) leads to a significant reduction in inhibitory activity against both kinases. This trend is also observed for the corresponding alcohol derivatives (compounds **2** and **4**). Furthermore, in line with our previous studies [28], we demonstrated that replacing the benzene ring in compound **F** (Figure 1) with a methyl group (as in compound **1**) enhances the inhibitory activity against both PIM-1 and CK2 kinases. Molecular docking and molecular dynamics (MD) calculations, combined with extended interaction mapping, provide structure-based insights into the molecular mechanisms that position compound **1** as a strong candidate for PIM-1 inhibition and a weak binder for CK2α. The anti-cancer activity of compound **1** is primarily attributed to its ability to induce apoptosis in leukemia and breast cancer cells via the intrinsic apoptotic pathway. The improved pro-apoptotic properties of compound **1** over the parent compound (TBBi) are likely due to its more efficient intracellular inhibition of PIM-1 rather than CK2. Moreover, our studies confirm the therapeutic relevance of simultaneously targeting PIM kinase and the NF-κB pathway in leukemic cells.

## Data Availability

All data are presented in the manuscript, and raw data (such as for immunoblots) will be available upon request.

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
