# Peer review of "Pro-Apoptotic Activity of 1-(4,5,6,7-Tetrabromo-1H-benzimidazol-1-yl)propan-2-one, an Intracellular Inhibitor of PIM-1 Kinase in Acute Lymphoblastic Leukemia and Breast Cancer Cells"

_ijms, 2025, doi:10.3390/ijms26125897_

Round 1

Reviewer 1 Report

Comments and Suggestions for Authors

The authors should explain what is novel as compared to their previous publications and why is this publication justified:

Bioorg Chem. 2024 Dec;153:107880. ; Pharmaceutics. 2023 Jul 20;15(7):1991.; Bioorg Chem. 2018 Oct;80:266-275.

I am a clinician so I wuld requet mor clinical relevance, especially that the paper should have more added value v. previous manuscripts.

The title suggests ALL and breats cancer specificity.

Are there any data for CK2 alpha and PIM-1 in ALL?

Nucleotide variants, structural variants, theire relations with disease features? If not published databases could be re-analysed. The same for expression.

Are there any trails of relevant inhibitors? Are the inhibitors (or genetic KD/KO) to be looked up in available cell lines screens such as DepMap?

What is the particular ALL relevance?

Are there any data for CK2 alpha and PIM-1 in breast cancer?

Nucleotide variants, structural variants, theire relations with disease features? If not published databases could be re-analysed. The same for expression.

Are there any trails of relevant inhibitors? Are the inhibitors (or genetic KD/KO) to be looked up in available cell lines screens such as DepMap?

What is the particular breast cancer relevance?

Why were the particular cell lines selected? What is the genomic and expression status of the targets? Versusu other cell lines? Why not e.g. HeLa?

Author Response

Comment 1: The authors should explain what is novel as compared to their previous publications and why is this publication justified:

Bioorg Chem. 2024 Dec;153:107880. ; Pharmaceutics. 2023 Jul 20;15(7):1991.; Bioorg Chem. 2018 Oct;80:266-275.

Response 1: Our previous studies described compounds that inhibited recombinant PIM kinase in vitro but lacked cellular activity. Here, for the first time, we report a TBBi derivative that effectively inhibits intracellular PIM kinase, with significantly greater selectivity for PIM over CK2 kinase. Regarding that CK2 kinase phosphorylates the p65 subunit of the NF-κB transcription factor, these findings are particularly noteworthy in light of the recent studies demonstrating the therapeutic potential of dual inhibition of the NF-κB pathway and PIM-1 kinase in ALL (Lim et al. Mol Cancer Ther. 2020 Sep;19(9):1809-1821. doi: 10.1158/1535-7163.MCT-20-0160)

To underline the relevance of this findings, we replaced the sentence in Conclusion “These findings further support the notion that PIM kinase plays a critical role in cancer cell survival and serves as an attractive molecular target for the development of novel anti-cancer agents.” into the following sentence: Moreover, our studies confirm the therapeutic relevance of simultaneously targeting PIM kinase and the NF-κB pathway in leukemic cells.”

Comment 2: I am a clinician so I wuld requet mor clinical relevance, especially that the paper should have more added value v. previous manuscripts. The title suggests ALL and breats cancer specificity. Are there any data for CK2 alpha and PIM-1 in ALL?

Response 2: Numerous studies have identified PIM kinases as potential therapeutic targets in T-cell acute lymphoblastic leukemia (T-ALL) [Lin et al., Blood 2010;115(4):824–33; 10.1182/blood-2009-07-233445; Padi et al., Oncotarget 2017;8(18):30199–216; 10.18632/oncotarget.16320; La Starza et al. Leukemia 2018;32(8):1807–10; 10.1038/s41375-018-0031-2; De Smedt et al. Haematologica 2019;104(1):e17–e20; 10.3324/haematol.2018.199257 ]. Other studies demonstrated that Tal-1 accelerates the development of T-ALL through the activation of protein kinase CK2 [Kelliher et al., Embo J. 1996;15(19):5160–6]. Furthermore, Silva A et al. showed that the combination of γ-secretase inhibitors and CK2 inhibitors holds therapeutic potential for T-ALL treatment [Silva et al., Haematologica. 2010;95(4):674–8; 10.3324/haematol.2009.011999]. Additionally, Padgaonkar A et al. developed a dual inhibitor, N108110, targeting both CK2 and CDK4/6, which effectively inhibited T-ALL progression [Padgaonkar A et al. Oncotarget. 2018;9(102):37753–65; doi.org/10.18632/oncotarget.26514].

Comment 3: Nucleotide variants, structural variants, theire relations with disease features? If not published databases could be re-analysed. The same for expression.

Response 3: In 1995, Seldin et al. identified a correlation between protein kinase CK2 and hematological malignancies. Their mouse model studies demonstrated that elevated expression of CSNK2A1 was linked to an increased risk of lymphoma [Seldin et al. Science. 1995;267(5199):894–7; 10.1126/science.7846532]. Since then, CK2 has been implicated in the pathogenesis of various other hematological cancers, including leukemia, and its activity has been associated with poorer clinical outcomes. These findings suggest that targeting protein kinase CK2 could represent a promising therapeutic strategy for the treatment of hematological malignancies.

Gene expression profiling using a publicly available T-ALL dataset revealed that PIM1 is overexpressed in the majority of early T-cell precursor (ETP)-ALL cases, as well as in a small subset of non-ETP ALL. Although PIM inhibitors effectively suppressed leukemic cell growth, they also induced phosphorylation of ERK and STAT5, indicating compensatory activation of alternative signaling pathways.

Comment 4: Are there any trails of relevant inhibitors? Are the inhibitors (or genetic KD/KO) to be looked up in available cell lines screens such as DepMap? What is the particular ALL relevance?

Response 4: Over the past two decades, numerous PIM kinase inhibitors have been developed; however, only a limited number have progressed to clinical trials. In a recent 2024 study, Chen et al. highlighted the therapeutic potential of PIM kinases as anticancer targets [S. Chen et al. Eur. J. Med. Chem. 264 (2024) 116016, https://doi.org/10.1016/J.EJMECH.2023.116016]. While several PIM inhibitors have demonstrated promising results in preclinical studies, including SGI-1776, GDC-0339, CX-6258, TP-3654, AZD1208, and PIM447, these agents have yet to show significant progress in clinical development. Many of them we can find in DepMap. To date, several PIM kinase inhibitors have entered clinical evaluation and demonstrated promising properties in patients [Chen et al. Eur. J. Med. Chem. 264 (2024) 116016, https://doi.org/10.1016/J.EJMECH.2023.116016.]. SGI-1776, an imidazo[1,2-b]pyridazine derivative, was the first PIM inhibitor to reach clinical trials for the treatment of relapsed and refractory leukemias [Chen et al. Blood 118 (2011) 693–702]. However, the trial was discontinued due to cardiotoxicity resulting from its lack of selectivity over the hERG potassium channel (ClinicalTrials.gov, NCT01239108). A subsequent effort led to the development of TP-3654 , another imidazo[1,2-b]pyridazine derivative. As a second-generation compound, TP-3654 shows improved selectivity against hERG and cytochrome P450 enzymes and is currently undergoing clinical evaluation (ClinicalTrials.gov, NCT03715504). CX-6258, an oxindole-based inhibitor identified through high-throughput screening, represents a distinct chemical class of PIM-1 inhibitors. It has demonstrated strong antiproliferative activity against the AML cell line MV4-11, as well as various other cancer cell lines. To date, CX-6258 has undergone clinical evaluation.

Given the critical role of CK2 in various hematological malignancies, inhibitors targeting this kinase have emerged as promising therapeutic candidates. One such compound, CX-4945 (Silmitasertib), is a selective CK2 inhibitor that induces apoptosis in tumor cells and enhances their sensitivity to chemotherapy by blocking CK2 activity and its downstream signaling pathways [Siddiqui-Jain et al. Cancer Res. 2010;70(24):10288–98; 10.1158/0008-5472.CAN-10-1893]. CX-4945 is currently undergoing clinical trials and has shown favorable anti-tumor activity, particularly in leukemia and multiple myeloma [Buontempo et al. Leukemia. 2017;32(1):1–10; 10.1038/leu.2017.301]. Another CK2-targeting agent in clinical trials is CIGB-300 [Perea et al. Semin Oncol. 2018;45(1–2):58–67; 10.1053/j.seminoncol.2018.04.006 ] which has also demonstrated encouraging therapeutic effects in hematological cancers [Rosales et al. Biomedicines. 2021. 10.3390/biomedicines9070766].

Comment 5: Are there any data for CK2 alpha and PIM-1 in breast cancer?

Response 5: Upregulation of PIM1 mRNA and protein expression has been observed in human breast cancer cells compared to normal breast tissue [Bray et al. CA Cancer J Clin. 2018 Nov;68(6):394-424. doi: 10.3322/caac.21492. Epub 2018 Sep 12. Erratum in: CA Cancer J Clin. 2020 Jul;70(4):313. doi: 10.3322/caac.21609]. It has been reported that elevated PIM-1 protein levels were associated with increased malignancy and higher tumor grade [Chen et al. Onco. Targets Ther. 12 (2019) 6267; 10.2147/OTT.S212752]. These findings suggest that high PIM-1 expression may serve as a negative prognostic indicator, particularly in patients with recurrent or treatment-resistant breast cancers.

Comment 6: Nucleotide variants, structural variants, theire relations with disease features? If not published databases could be re-analysed. The same for expression.

Response 6: The research demonstrated that human breast cancer tissues exhibit elevated CK2 catalytic activity, often correlating with CK2 overexpression, thereby suggesting a pathological link between CK2 expression and mammary tumorigenesis [Landesman-Bollag E et al., Oncogene. 2001;20(25):3247–3257; doi: 10.1038/sj.onc.1204411]. Given the relevance of molecular subtype classification in breast cancer prognosis and treatment, Giusiano et al. investigated the prognostic significance of CK2α expression in relation to clinicopathological parameters across a large breast tumor cohort. Their findings revealed a strong association between CK2α overexpression and aggressive tumor phenotypes, positioning CK2 as a negative prognostic marker [Giusiano et al. Eur J Cancer. 2011;47(5):792–801; 10.1016/j.ejca.2010.11.028]. At the transcriptomic level, both CK2α and CK2β are significantly upregulated in breast tumors and correlate with poor survival outcomes [Gray et al., Oncotarget. 2014;5(15):6484–6496; 10.18632/oncotarget.2248; Ortega et al., PLoS ONE. 2014;9(12):e115609; 10.1371/journal.pone.0115609]. This aligns with two independent studies that identified CK2 as part of an "invasiveness gene signature" predictive of metastasis and reduced survival in breast cancer patients [Liu et al., N Engl J Med. 2007;356(3):217–226;  10.1056/NEJMoa063994]. Clinically, breast tumors are commonly classified into three therapeutic subtypes: estrogen receptor-positive (ER+), HER2-overexpressing, and triple-negative breast cancers (TNBCs), which lack ER, PR, and HER2 expression. A recent transcriptomic analysis revealed consistent overexpression of CK2α and CK2β, alongside significant underexpression of CK2α′ across all subtypes—patterns that were associated with poorer survival outcomes [Ortega et al., PLoS ONE. 2014;9(12):e115609;  10.1371/journal.pone.0115609].

Although the relationship between transcript-level dysregulation and protein expression remains to be fully elucidated, these findings suggest that altered CK2 gene expression may contribute to the elevated CK2 activity and protein levels observed in specific breast cancer subtypes.

The oncogene PIM-1 has been found to be upregulated in breast cancer, particularly in triple-negative breast cancer (TNBC) [Chen et al., Onco Targets Ther. 2019;12:6267–6273. doi: 10.2147/OTT.S212752]. It plays a role in tumorigenesis, contributes to the development of drug resistance, and is associated with poor clinical outcomes. Recently, the development of a highly selective imaging probe targeting PIM-1 has highlighted its potential as a biomarker for accurate diagnosis and targeted therapy in TNBC. The oncogene PIM-1 has been found to be upregulated in breast cancer, particularly in triple-negative breast cancer (TNBC). It plays a role in tumorigenesis, contributes to the development of drug resistance, and is associated with poor clinical outcomes. Recently, the development of a highly selective imaging probe targeting PIM-1 has highlighted its potential as a biomarker for accurate diagnosis and targeted therapy in TNBC.

Comment 7: Are there any trails of relevant inhibitors? Are the inhibitors (or genetic KD/KO) to be looked up in available cell lines screens such as DepMap? What is the particular breast cancer relevance?

Response 7: Currently, there are no known clinical trials specifically investigating PIM kinase inhibitors for the treatment of breast cancer. However, several preclinical studies suggest promising therapeutic potential, particularly in triple-negative breast cancer (TNBC) and tumors with MYC overexpression.

Preclinical studies have shown that inhibition of PIM1 in MYC-overexpressing TNBC models suppressed tumor growth by restoring the activity of cell cycle inhibitor p27 and disrupting MYC-driven transcriptional programs [Horiuchi et al., Nat Med. 2016 Nov;22(11):1321-1329; 10.1038/nm.4213]. Other studies have shown co-inhibition of PIM kinases and the proteasome exhibited synergistic anti-tumor effects in MYC-high TNBC. The mechanism involved accumulation of polyubiquitinated proteins, proteotoxic stress, and impaired NRF1 function [Kunder et al., Cell Chem Biol. 2022 Mar 17;29(3):358-372.e5. doi: 10.1016/j.chembiol.2021.08.011]. In other preclinical studies, it has been demonstrated that IBL-302 showed efficacy in both in vitro and in vivo models by inhibiting pAKT, pmTOR, and pBAD signaling in various breast cancer cell lines, highlighting its multi-pathway targeting potential [Kennedy et al., Oncogene. 2020 Apr;39(14):3028-3040. doi: 10.1038/s41388-020-1202-y].

The ATP-competitive CK2 inhibitor CX-4945 has demonstrated promising outcomes in preclinical studies, indicating its potential as a therapeutic agent in breast cancer. In addition, CIGB-300 and K164 have exhibited anti-metastatic and cytotoxic effects in preclinical models, supporting the rationale for further clinical investigation. Nonetheless, all of these compounds require additional clinical trials to comprehensively assess their efficacy and safety in the context of breast cancer treatment.

Taking into account all of the above suggestions as well as the coherence of the introduction, we have added the following sections to the introduction (marked in yellow in the manuscript) with the relevant references:

Noteworthy, numerous  studies have identified PIM kinases as potential thera-peutic targets in T-cell acute lymphoblastic leukemia (T-ALL) [7,8] as well as in human breast cancer  [9]. Gene expression profiling using a publicly available T-ALL dataset revealed that PIM-1 is overexpressed in the majority of early T-cell precursor (ETP)-ALL cases, as well as in a small subset of non-ETP ALL. Upregulation of PIM-1 mRNA and protein expression has been also observed in human breast cancer cells compared to normal breast tissue [9]. It has been reported that elevated PIM-1 protein levels were associated with increased malignancy and higher tumor grade. These findings suggest that high PIM-1 expression may serve as a negative prognostic indi-cator, particularly in patients with recurrent or treatment-resistant breast cancers. In a recent 2024 study, Chen  et al. highlighted the therapeutic potential of PIM ki-nases as anticancer targets [5]. While several PIM inhibitors have demonstrated promising results in preclinical studies, including SGI-1776, GDC-0339, CX-6258, TP-3654, AZD1208, and PIM447, these agents have yet to show significant progress in clinical development [5].”

To highlight the importance of CK2 in ALL and breast cancer, we have also added the following sections in the introduction:

“Importantly, CK2 has been implicated in the pathogenesis of various hematological cancers, including leukemia  [13] as well as breast cancer  [14] and its activity has been associated with poorer clinical outcomes.”

“Another CK2-targeting agent in clinical trials is CIGB-300, which has also demonstrated encouraging therapeutic effects in hematological cancers as well as breast cancer  [18].”

Comment 8: Why were the particular cell lines selected? What is the genomic and expression status of the targets? Versusu other cell lines? Why not e.g. HeLa?

Response 8: Our research continues a long-standing investigation into the anticancer potential of TBBi derivatives. Both our previous studies and those conducted by other research groups have consistently shown that certain cell lines, particularly MCF-7 and CCRF-CEM, exhibit heightened sensitivity to these compounds.  This is likely due to the fact that the derivatives we have developed are potential inhibitors of CK2 kinase, whose overexpression has been reported in various cancers, including leukemia and breast cancer.

Reviewer 2 Report

Comments and Suggestions for Authors

The plagiarism needs to be reduced before considering the article for further processing 

Author Response

Comment: The plagiarism needs to be reduced before considering the article for further processing 

Response: We corrected manuscript to reduce the plagiarism. Revised sections are linked in green in the manuscript.

Reviewer 3 Report

Comments and Suggestions for Authors

In this work, the authors synthesized derivatives of 4,5,6,7-tetra-13-bromo-1H-benzimidazole and characterized them using different cell lines to evaluate their anti-cancer activity. Results show intracellular inhibition of CK2 and PIM-1 kinases in CCRF-CEM and MCF-7 cells. One of the synthesized compounds was revealed to be interesting, since it leads to pro-apoptotic activity and the ability to inhibit PIM-1 kinase intracellularly, with possible application in leukemia treatment. The manuscript is clear and detailed, the experiments and the results are well described, and the conclusions follow the results, therefore, it can be considered for publication, if the following minor revision is considered:

  • Please increase the size of the numbers on the graphs' scales in Figures 3 (c), 4(c), 5(b), 5(d),  6(b),  6(d), 7(b), AND 9(b) to become readable, and also increase the size of the numbers and the letters indicating the atoms in the chemical structures in Figures 11 (b) and 12(c).
  • Please include error bars in the values of Tables 1 and 2.
  • If possible compare the results achieved with the TBBi with those described in the literature and achieved with other molecules for the same purpose.

Author Response

In this work, the authors synthesized derivatives of 4,5,6,7-tetra-13-bromo-1H-benzimidazole and characterized them using different cell lines to evaluate their anti-cancer activity. Results show intracellular inhibition of CK2 and PIM-1 kinases in CCRF-CEM and MCF-7 cells. One of the synthesized compounds was revealed to be interesting, since it leads to pro-apoptotic activity and the ability to inhibit PIM-1 kinase intracellularly, with possible application in leukemia treatment. The manuscript is clear and detailed, the experiments and the results are well described, and the conclusions follow the results, therefore, it can be considered for publication, if the following minor revision is considered:

Comment 1: Please increase the size of the numbers on the graphs' scales in Figures 3 (c), 4(c), 5(b), 5(d),  6(b),  6(d), 7(b), AND 9(b) to become readable, and also increase the size of the numbers and the letters indicating the atoms in the chemical structures in Figures 11 (b) and 12(c).

Response 1: We have corrected the following figures: Figures 11(b) and 12(b), which were highlighted in blue in the manuscript. However, we are unable to modify the original figures from the BD Accuri C6 Plus software, as the software does not permit such alterations.

Comment 2: Please include error bars in the values of Tables 1 and 2.

Response 2: Standard deviations (SD) have been added to the values in Tables 1 and 2, highlighted in red in the manuscript.

Comment 3: If possible compare the results achieved with the TBBi with those described in the literature and achieved with other molecules for the same purpose.

Response 3: The results obtained for TBBi describing cellular viability of the tested cells are comparable with those obtained so far. For example, in our previous paper [Łukowska-Chojnacka et al., Bioorg Chem. 2024, 153, 107880. https://doi.org/10.1016/j.bioorg.2024.107880, reference 34 in the manuscript], IC50 values for CCRF-CEM and MCF-7 were 17.18 µM and 14.37 µM, respectively, whereas in the present paper these values are: 17.56±0.67 µM for CCRF-CEM and 12.61±2.72 for MCF-7. In another paper [Chojnacki et al., Int J Mol Sci. 2021 Jun 10;22(12):6261. doi: 10.3390/ijms22126261, reference 33 in the manuscript], these values were 18.82 ± 1.60 µM for MCF-7 and 25.11 ± 2.04 µM for MDA-MB-231, whereas in the present paper IC50 for TBBi for MDA-MB-231 is 29.71±2.23 µM.

We added the following sentence with the references into the Results: “The results obtained for TBBi describing cellular viability of the tested cells are comparable with those obtained so far [33,34].” (grey color).

The results describing the inhibition of recombinant kinases can vary depending on the methods and reaction conditions used. Since TBBi is an ATP-competitive inhibitor of CK2 and PIM kinase, the observed IC50 values may differ based on ATP concentration and/or the specific assay employed. However, IC50 values for CK2 inhibition by TBBi are usually in the range of 0.5 to 1.5 µM  [Leung et al., Biochemistry 2015 54 (1), 47-59; DOI: 10.1021/bi500959t; Chojnacki et al., Bioorg Chem. 2021 Jan;106:104502. doi: 10.1016/j.bioorg.2020.104502; Najda-Bernatowicz et al., Bioorg Med Chem. 2009 Feb 15;17(4):1573-8. doi: 10.1016/j.bmc.2008.12.071; Bretner et al., Mol Cell Biochem. 2008 Sep;316(1-2):87-9. doi: 10.1007/s11010-008-9827-0].

The other results related to the induction of apoptosis, mitochondrial potential, cell cycle progression, and western blot analysis are influenced by the doses of TBBi used. For example in our previous paper [Łukowska-Chojnacka et al., Bioorg Chem. 2024, 153, 107880. https://doi.org/10.1016/j.bioorg.2024.107880], these doses were 5 µM and 10 µM for western blot analysis. Therefore, the obtained effects can be different.

Reviewer 4 Report

Comments and Suggestions for Authors

Dear Authors,

I have read with due diligence the manuscript entitled “Pro-apoptotic activity of 1-(4,5,6,7-tetrabromo-1H-benzimidazol-1-yl)propan-2-one, an intracellular inhibitor of PIM-1 kinase in acute lymphoblastic leukemia and breast cancer cells” by Patrycja Wińska et al., and I hereby inform you that I accept the article for publication in Molecular Science, provided that the authors address the comments I have categorized as minor revisions.

I kindly ask the authors to respond to the following remarks.

My attention was drawn to certain - let us call them - shortcomings, though not outright errors, particularly in the Conclusions section. For instance, the claim regarding an increase in the distance between the carbonyl group and the TBBi scaffold is somewhat problematic. Such a conclusion would be plausible if the system in question were rigid - say, incorporated into a heterocyclic framework. However, the introduced CH₂ carbon (sp³ hybridized), adjacent to another carbon of similar hybridization, increases the rotational freedom around these bonds. In other words, due to this conformational flexibility, the side chain may fold toward the bicyclic system. Thus, while an increased distance is plausible, it is by no means unambiguous. A more convincing interpretation would require a deeper discussion of the nature of the observed phenomenon.

I find myself missing an analysis in the spirit of the authors’ own language - particularly that used in line 785, where it is stated that the study provides “insight into structure-based molecular mechanisms”. This is an excellent and promising direction.

Therefore, in discussing the substitution of the aromatic ring with a methyl group, which enhances the inhibitory activity of the synthesized compounds toward PIM-1 and CK2 kinases, it is worth emphasizing the electronic consequences of such a structural modification. Specifically, when a carbonyl group is directly attached to an aromatic ring, π-conjugation occurs between the carbonyl and the ring system. The π-electrons of the C=O bond may delocalize onto the aromatic ring, and vice versa. As a result, the electron density on the carbonyl carbon decreases, rendering the group less electrophilic. I would strongly encourage the authors to consider incorporating a similar level of interpretation in their conclusions.

Kind regards,

Author Response

I have read with due diligence the manuscript entitled “Pro-apoptotic activity of 1-(4,5,6,7-tetrabromo-1H-benzimidazol-1-yl)propan-2-one, an intracellular inhibitor of PIM-1 kinase in acute lymphoblastic leukemia and breast cancer cells” by Patrycja Wińska et al., and I hereby inform you that I accept the article for publication in Molecular Science, provided that the authors address the comments I have categorized as minor revisions.

I kindly ask the authors to respond to the following remarks.

Comment 1: My attention was drawn to certain - let us call them - shortcomings, though not outright errors, particularly in the Conclusions section. For instance, the claim regarding an increase in the distance between the carbonyl group and the TBBi scaffold is somewhat problematic. Such a conclusion would be plausible if the system in question were rigid - say, incorporated into a heterocyclic framework. However, the introduced CH₂ carbon (sp³ hybridized), adjacent to another carbon of similar hybridization, increases the rotational freedom around these bonds. In other words, due to this conformational flexibility, the side chain may fold toward the bicyclic system. Thus, while an increased distance is plausible, it is by no means unambiguous. A more convincing interpretation would require a deeper discussion of the nature of the observed phenomenon.

Response 1: We agree that the statement in the summary is incorrect and that the introduction of the CH2 group does not necessarily increase the distance between the carbonyl group and the TBBi scaffold. We have made the appropriate changes to the manuscript to correct this. In the “Introduction” section the sentence:

 “We decided to investigate how the replacement of the benzene ring with a small methyl substituent and a slight increase in the distance between the TBBi scaffold and the carbonyl group would affect the anti-cancer properties and affinity of the compounds for CK2 and PIM-1.”

has been revised to:

 „We decided to investigate how the replacement of the benzene ring with a small methyl substituent and a slight modification of the alkyl chain between the TBBi scaffold and the carbonyl group would affect the anti-cancer properties and affinity of the compounds for CK2 and PIM-1.”

Corresponding changes have also been made in the “Conclusion” section. The sentence:

“Increasing the distance between the TBBi scaffold and the carbonyl group by one CH2 group (i.e., compound 1 vs. compound 3) leads to a significant reduction in inhibitory activity against both kinases. This trend is also observed for the corresponding alcohol derivatives (compounds 2 and 4).”

has been changed to:

“The insertion of an additional CH2 group between the benzimidazole scaffold and carbonyl group (i.e., compound 1 vs. compound 3) leads to a significant reduction in inhibitory activity against both kinases. This trend is also observed for the corresponding alcohol derivatives (compounds 2 and 4)."

Comment 2: I find myself missing an analysis in the spirit of the authors’ own language - particularly that used in line 785, where it is stated that the study provides “insight into structure-based molecular mechanisms”. This is an excellent and promising direction.

Therefore, in discussing the substitution of the aromatic ring with a methyl group, which enhances the inhibitory activity of the synthesized compounds toward PIM-1 and CK2 kinases, it is worth emphasizing the electronic consequences of such a structural modification. Specifically, when a carbonyl group is directly attached to an aromatic ring, π-conjugation occurs between the carbonyl and the ring system. The π-electrons of the C=O bond may delocalize onto the aromatic ring, and vice versa. As a result, the electron density on the carbonyl carbon decreases, rendering the group less electrophilic. I would strongly encourage the authors to consider incorporating a similar level of interpretation in their conclusions.

Response 2: In our series, the carbonyl group is present only in compounds 1 and 3, and it resides in the side chain, not directly connected to the ring system. The side chain in our compounds is relatively flexible and can adopt multiple conformations to fit within the active site of the target kinases. Empirically, we observed that a shorter linker containing a carbonyl group (rather than a hydroxyl) provides a better fit, which was supported by molecular dynamics (MD) simulations. Our MD results indicate that such a structural motif enables more favorable enthalpic interactions within the active sites of both CK2 and PIM-1 kinases. While we recognize that electronic effects might play an important role in binding affinity, we refrained from drawing conclusions on their influence, as MD simulations do not adequately capture these effects.

Round 2

Reviewer 1 Report

Comments and Suggestions for Authors

I acknowledge the authors' response to my queries and their extensions to the manuscipt.

The manuscipt is now satisfactory for publication.